

# THAPBI PICT—a fast, cautious, and accurate metabarcoding analysis pipeline

Peter J. A. Cock[1], David E. L. Cooke[2], Peter Thorpe[2,3] and
Leighton Pritchard[1,4]

[1] Information and Computational Sciences, The James Hutton Institute, Dundee, United Kingdom
[2] Cell and Molecular Sciences, The James Hutton Institute, Dundee, United Kingdom
[3] The Data Analysis Group, School of Life Sciences, The University of Dundee, Dundee, United Kingdom
[4] Strathclyde Institute of Pharmacy and Biomedical Sciences, University of Strathclyde, Glasgow, United Kingdom

Corresponding author
Peter J. A. Cock,
peter.cock@hutton.ac.uk

## ABSTRACT

THAPBI PICT is an open source software pipeline for metabarcoding analysis of Illumina paired-end reads, including cases of multiplexing where more than one amplicon is amplified per DNA sample. Initially a *Phytophthora* ITS1 Classification Tool (PICT), we demonstrate using worked examples with our own and public data sets how, with appropriate primer settings and a custom database, it can be applied to other amplicons and organisms, and used for reanalysis of existing datasets. The core dataflow of the implementation is (i) data reduction to unique marker sequences, often called amplicon sequence variants (ASVs), (ii) dynamic thresholds for discarding low abundance sequences to remove noise and artifacts (rather than error correction by default), before (iii) classification using a curated reference database. The default classifier assigns a label to each query sequence based on a database match that is either perfect, or a single base pair edit away (substitution, deletion or insertion). Abundance thresholds for inclusion can be set by the user or automatically using per-batch negative or synthetic control samples. Output is designed for practical interpretation by non-specialists and includes a read report (ASVs with classification and counts per sample), sample report (samples with counts per species classification), and a topological graph of ASVs as nodes with short edit distances as edges. Source code available from https://github.com/peterjc/thapbi-pict/ with documentation including installation instructions.

## INTRODUCTION

Metabarcoding of DNA is a sensitive and powerful method to detect, identify, and potentially quantify the diversity of biological taxa present in any given environmental sample. It is based on PCR amplification of a "barcode" region diagnostic for the groups of organisms of interest followed by high-throughput sequencing of the amplimers, and is often applied to environmental DNA (eDNA) samples (*Deiner et al., 2017*). This method is revolutionising areas of research including wildlife conservation, ecological processes and microbiology, by highly-sensitive detection of biodiversity across many taxa

simultaneously (*Arulandhu et al., 2017*). Metabarcoding enables early detection of invasive threats to plant and human health in support of biosecurity (*Batovska et al., 2021*; *Green et al., 2021*), and is applicable to many complex and intractable systems, such as soil (*Ahmed et al., 2019*), in which standard methods of microbial isolation and characterisation are impractical or costly.

Our motivating use case is metabarcoding in which multiple environmental samples are multiplexed for high-throughput sequencing on the Illumina platform using paired-end reads, and for which the expected PCR amplification product is short enough to be fully covered by the overlapping paired reads. Each sample is expected to yield taxon-specific marker sequences that can be matched to a high-quality database of marker sequences with known taxonomic identity, to give a taxonomic breakdown reflecting the community composition. One of our goals was to minimise false positive identification of the presence of any taxon on the basis of small or disputable quantities of physical evidence. Metabarcoding is prone to generation of artefactual sequence variation and sufficiently highly sensitive to register low-abundance sample reads at the same level as such sequences, and sequences originating from cross-sample contamination and "splashover" in even a careful laboratory. We therefore chose to prioritise accurate reporting of taxonomic assignment for high abundance sequences over sensitive detection of low-abundance marker sequences.

This manuscript was initially released as a preprint (*Cock et al., 2023*). We describe THAPBI PICT v1.0.0, a metabarcoding tool developed as part of the UKRI-funded Tree Health and Plant Biosecurity Initiative (THAPBI) Phyto-Threats project, which focused on identifying *Phytophthora* species in commercial forestry and horticultural plant nurseries (*Green et al., 2021*). *Phytophthora* (from Greek meaning plant-destroyer) is an economically important genus of oomycete plant pathogens that causes severe losses and damage to plants in agricultural, forest and natural ecosystems. The Phyto-Threats project's metabarcoding protocol used nested PCR primers designed to target the Internal Transcribed Spacer 1 marker sequence (ITS1; a genomic region located between 18S and 5.8S rRNA genes in eukaryotes) of *Phytophthora* and related plant pathogenic oomycetes (*Scibetta et al., 2012*). This approach is the current *de facto* standard within the oomycete community (*Robideau et al., 2011*), and these primers have been used in conjunction with THAPBI PICT in recent *Phytophthora* surveys (*Vélez et al., 2020*; *La Spada et al., 2022*). PICT was short for *Phytophthora* ITS1 Classification Tool.

We describe the implementation, operation, performance and output of THAPBI PICT using datasets from the Phyto-Threats project, and public metabarcoding datasets. Although originally designed as a *Phytophthora* ITS1 Classification Tool (PICT), we show that with appropriate primer settings and a custom database of genus/species distinguishing markers, THAPBI PICT is an effective tool for analysis of short read amplicon sequencing data with barcode marker sequences from other organisms.

## WORKFLOW OVERVIEW

The THAPBI PICT core workflow comprises (i) data reduction to unique marker sequences, often called amplicon sequence variants (ASVs) (ii) discard of low abundance

sequences to remove noise and artifacts (rather than attempting error correction by default), and (iii) classification using a curated reference database. This approach contrasts with commonly-used operational taxonomic unit (OTU) clustering approaches (as implemented, for example, in QIIME (*Caporaso et al., 2010*), UPARSE (*Edgar, 2013*), and MOTHUR (*Schloss et al., 2009*)), which can be sensitive to changes in the input data resulting in unpredictable clustering behaviour (*Callahan, McMurdie & Holmes, 2017*) and overestimate population diversity (*Nearing et al., 2018*).

THAPBI PICT's approach of reducing amplicons to ASVs is similar to that of DADA2 (*Callahan et al., 2016*) but, by contrast, THAPBI PICT does not by default attempt to correct sequencing errors with a denoising model. Our approach is instead to discard low-abundance sequences because we consider that they are likely not to represent meaningful biological information in the sequenced sample. We observe, using synthetic control sequences, that the abundance of such controls accidentally transferred between samples tends to exceed by no small margin the abundance of amplicons whose sequence variation might constitute "noise" in the amplicon sequence data. We consider the observed abundance of (*e.g.*, synthetic) control sequences, which could not have been present in the biological sample, to be a lower bound for the abundance of reads we can confidently claim derive from that sample. Consequently, ASVs with much lower total abundance cannot confidently be determined to derive from the analysed sample, and so are discarded. In general, we consider that proper use of negative and synthetic controls, to account for alternative sources of experimental error, such as accidental transfer or "splashover" from one well to another, should be considered best practice in metabarcoding.

Figure 1 gives an overview of the workflow. Paired raw Illumina FASTQ files for each sample are merged by overlap, trimmed to remove primers, and reduced to a list of observed unique marker sequences (labelled by MD5 checksum) with abundance counts. Discarding low abundance sequences further reduces the data volume; unique reads alone may represent half the data (and 90% of the ASVs), but may not derive from the sequenced sample. The remaining higher abundance sequences are then classified by matching them to a curated database. By default a species-level assignment is made when a database entry is identical or different by at most one base pair (1 bp; algorithm `onebp`) to the query. The matching algorithm can be chosen to adjust sensitivity for taxonomic classification (Table 1).

Following read preparation and ASV classification, the pipeline generates two tables describing (i) taxon presence/absence for each sample, and (ii) ASV presence/absence for each sample (Figs. 2A and 2B respectively), in both plain text and Excel format. If the user provides suitably formatted sample metadata, cross-referenced by the filename stem, this can be incorporated into the report to make for easier interpretation. Additionally, an edit-graph showing the distances between the ASVs recorded in the sample can be exported (*e.g.*, Fig. 3).
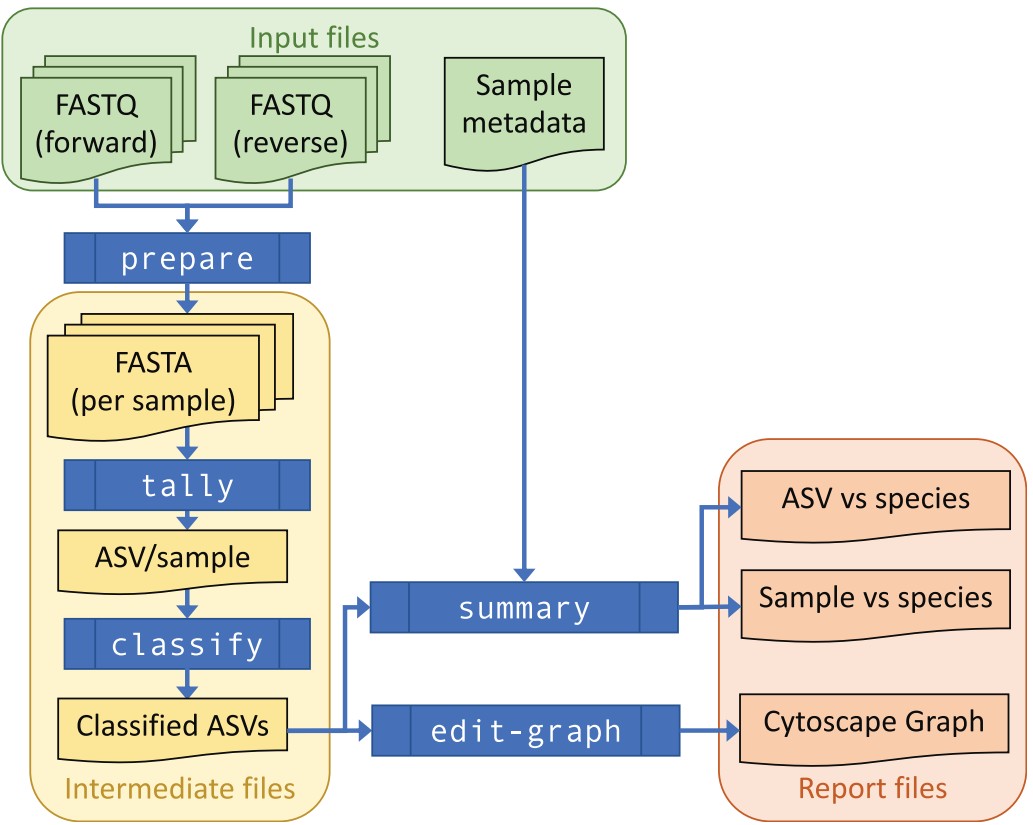

**Figure 1** **THAPBI PICT workflow overview.** Raw paired FASTQ input data is transformed (commands `prepare-reads`, `sample-tally`, `classify`) into intermediate FASTA and TSV (tab-separated value) format files recording tallies of ASV counts and ASV classifications, using a local marker sequence database. Summary report generation (command `summary`) produces output in reproducible (TSV, TXT file) and user-focused colour-coded Excel spreadsheet formats. The stages of THAPBI PICT can be run individually, or as a single `pipeline` command that incorporates the `prepare-reads`, `sample-tally`, `classify` and `summary` commands. Sample metadata can optionally be incorporated into report output, and used to sort reports and support downstream interpretation. In addition, Biological Observation Matrix (BIOM) format output can be requested. An ASV edit graph for the samples can be generated (command `edit-graph`) to aid in diagnosis and interpretation.

**Table 1** **Taxonomic classifier algorithms in THAPBI PICT.** Names constructed as `XsYg` reflect an edit distance of up to and including `X`bp for species classification, and `Y`bp for genus-level classification. Genus-level classification does not attempt to assign a species-level identity to the sequence.

| Name | Description |
| --- | --- |
| identity | Perfect match in database (strictest) |
| substr | Perfect match or perfect substring of a database entry |
| onebp | Perfect match, or one bp away (default) |
| 1s2g | As onebp but falling back on up to 2 bp away for a genus only match. |
| 1s3g | As onebp but falling back on up to 3 bp away for a genus only match. |
| 1s4g | As onebp but falling back on up to 4 bp away for a genus only match. |
| 1s5g | As onebp but falling back on up to 5 bp away for a genus only match. |
| blast | Best NCBI blastn alignment covering at least 85% of the query, and 95% identity. |

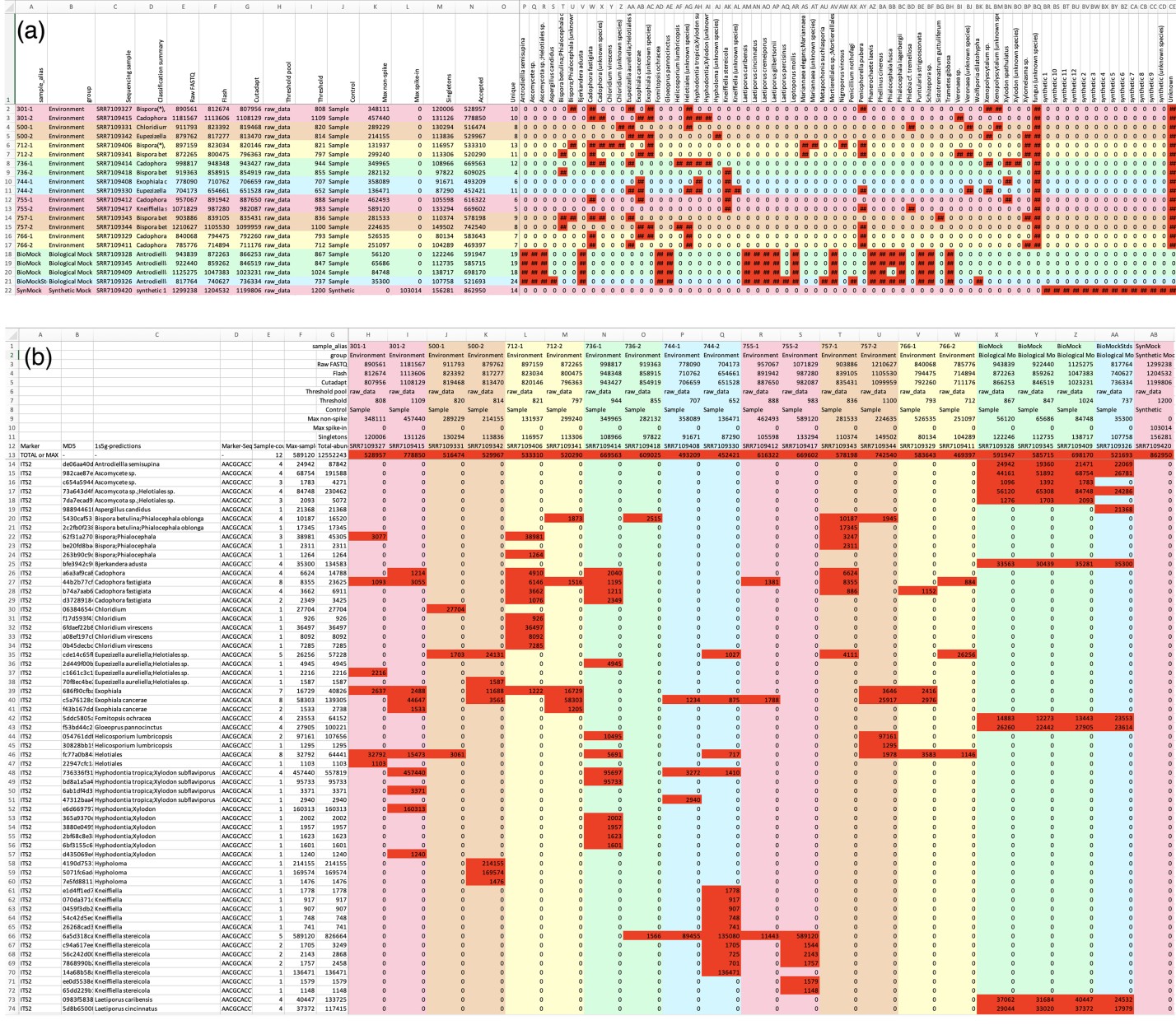

**Figure 2** Screenshots of the (A) sample and (B) read reports, using the "m6" ITS2 MiSeq run from *Palmer et al. (2018)*, also used in Figs. 3 and 4. Both tables show cells with read counts in the lower right section, using conditional formatting to apply a red background for non-zero entries. Excel shows read counts as "##" where the count is too wide for the column width, as in (A) where the default sample report layout prioritises showing an overview. The column widths in the sample report have been adjusted for display in (B), and the bottom of the table cropped. In this example two fields of user-supplied metadata (sample *alias* and group) are included in both reports, which have been used for sample sorting and the automatic use of a rainbow of five pastel background colours to visually show the sample groupings. In this case the environmental samples are in pairs. The next fields are from the data itself, reads counts in the samples as raw FASTQ, after read merging with Flash, primer trimming with Cutadapt, the abundance threshold applied, the maximum ASV read count for non-spike-in or spike-in sequences, number of singletons, number of unique ASVs accepted, and the total number of reads for the accepted ASVs. These fields were used to generate Fig. 4. The read report also includes the full ASV sequence and its MD5 checksum which is used internally as an identifier, and a concatenation of all the species present in the classifier output as a single field.

(a) Default 0.1% abundance threshold, showing 360 ASVs:

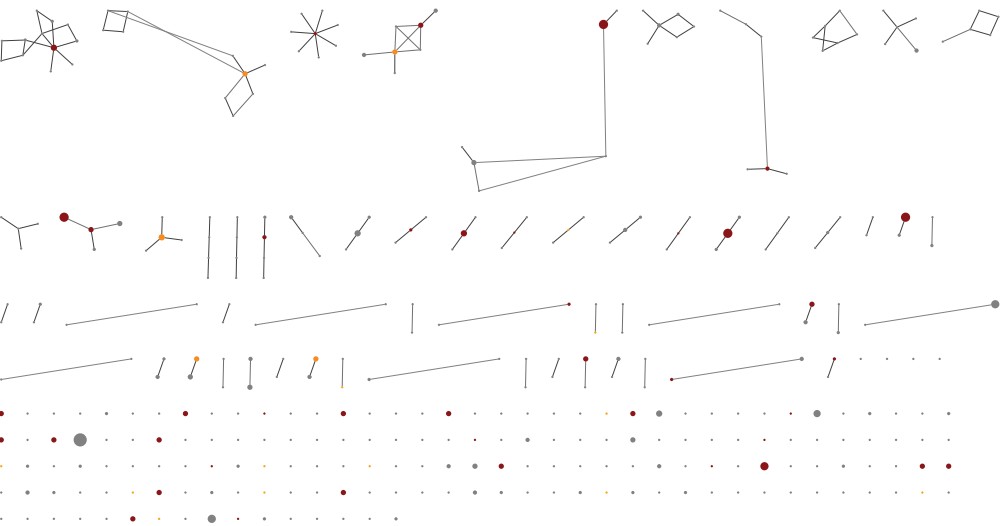

(b) Synthetic control inferred 0.0156% abundance threshold, showing 3097 ASVs:

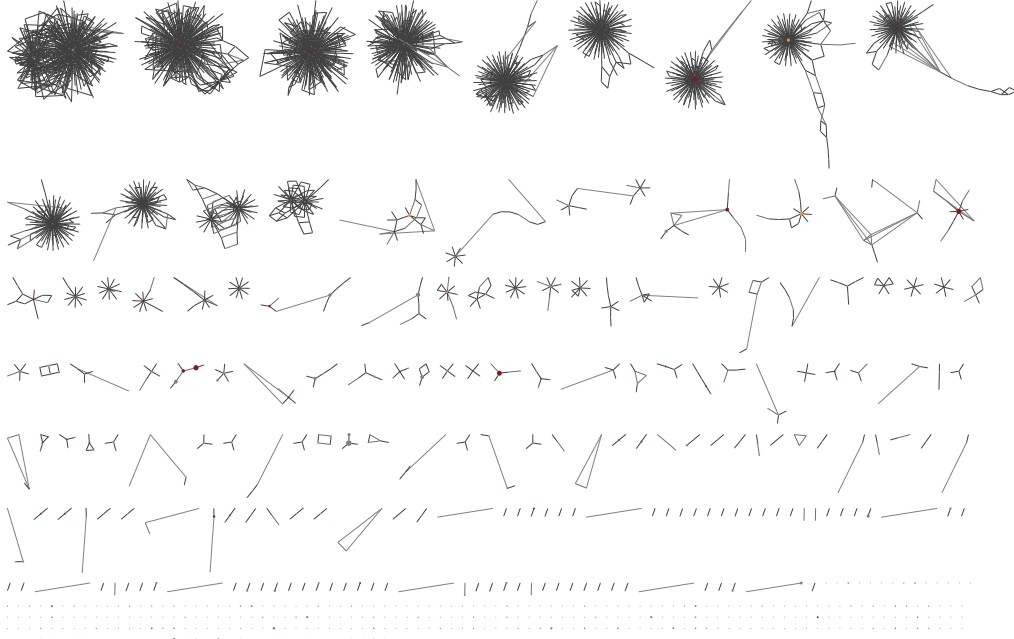

**Figure 3** **Example ASV edit-graph, exported as an XGMML format graph, then opened in Cytoscape v3.9.1 (*Shannon et al., 2003*) showing ITS2 sequences from the same *Palmer et al. (2018)* MiSeq run shown in Figs. 2 and 4.** Each node represents an ASV, orange if matched to a synthetic control, dark red for a matched genus, grey otherwise. The node circles are scaled according to the number of samples it was in, and shown here without labels for clarity. The edges are solid for a one base pair edit distance, dashed for a two base pair edit distance, and dotted for a three base pair edit distance. The nodes were arranged in CytoScape using edge weighted prefuse force directed layout, and their placement is not consistent between (A) and (B). As the abundance threshold is lowered from (A) to (B), the number of nodes increases roughly ten-fold. The more common ASV nodes become the centre of a halo of 1 bp variants, typically each seen in a single sample, which we attribute to PCR noise and/or sequencing error.               

## Read preparation

The first and slowest stage of the workflow is read preparation. Paired raw Illumina FASTQ files are processed into intermediate FASTA files per amplicon marker containing the ASV sequences and their abundances. It is simplest to run the pipeline on all input data sequentially, but with large projects or for most efficient usage of a computer cluster it is advisable to run the read preparation step in batches, for example by MiSeq plate or sample batch, as separate jobs.

The first step is merging the overlapping FASTQ read pairs, currently done using Flash (*Magoč & Salzberg, 2011*). This is invoked with the allow "outies" option and maximum overlap increased from the default 65 to 300 bp, which was especially important when working with smaller fragments. Initially we used Pear (*Zhang et al., 2014*), but open source development ended with Pear v 0.9.6, and Flash was faster with equivalent output. The merged sequences for each sample are tallied (discarding the per-base quality scores), which avoids re-processing repeated sequences in each sample.

Next, we use cutadapt (*Martin, 2011*) to identify each amplicon sequence using the primer sequences, which are then removed. These shorter unique sequences in each sample are re-tallied, and unique reads appearing only once in a sample (singletons) are discarded at this point. This gives a list of ASVs with counts per marker per sample.

Earlier versions of the tool and the pre-cursor `metapy` pipeline (*Riddell et al., 2019*) started by removing the Illumina adapter sequences using Trimmomatic (*Bolger, Lohse & Usadel, 2014*), before merging the overlapping reads. Flash was developed before tools like Trimmomatic, and does not require this. Skipping adapter trimming at the start was faster, and made minimal difference to the output, especially since any residual adapter sequence is removed when primer trimming.

Collectively our dataset for the Phyto-Threats project (*Green et al., 2021*) and related work including natural ecosystems (*Riddell et al., 2019*), is now over 30 MiSeq plates, with several thousand sequenced samples. To balance performance *vs* complexity we run the read-preparation by plate. In a typical run on HPC nodes with 2nd-Gen Xeon Scalable (Cascade Lake; 2019) processors preparing the slowest plate took 12.5 min, while global tallying through to reporting (see below) added a further 7.5 min, giving a total elapsed time of approximately 20 min.

## Sample tallying and optional read-correction

Once all the FASTQ sample files have been prepared (which is the slowest part of the pipeline), the unique ASVs are tallied per marker per sample. This workflow accommodates large projects where new plates of MiSeq data are sequenced over time, and exploring the effect of adjusting settings like the abundance thresholds.

At this point, before applying abundance thresholds (see below), optional read-correction can be applied. This can use our re-implementation of the original UNOISE2 read-correction method as described in *Edgar (2016)* using the Levenshtein distance as implemented in the Rapid Fuzz library (*Bachmann et al., 2022*). Alternatively, it can call the later UNOISE3 algorithm *via* Edgar's command line tool `usearch`, or as reverse engineered in `vsearch` (*Rognes et al., 2016*).

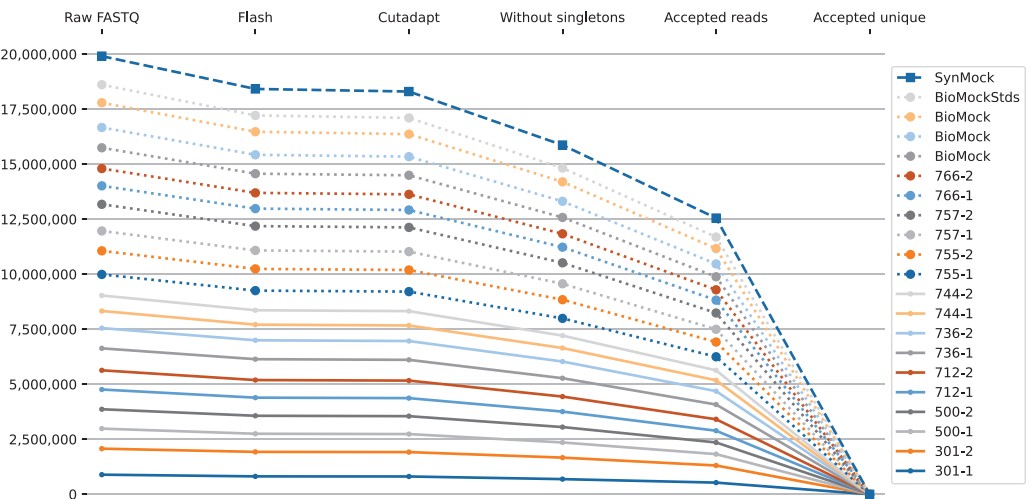

**Figure 4** Stacked line graph illustration of how the raw FASTQ read counts are reduced to ASV tallies, showing reads counts from ITS2 sequences from the same *Palmer et al. (2018)* MiSeq run shown in **Figs. 2** and **3**. Starting from raw FASTQ files, over 90% could be merged into overlapping reads, most of which could be primer trimmed. At this point the data is already held as ASV tally tables internally. The next drop represents removing singletons, leaving about 80% of the starting reads. Applying the default minimum abundance thresholds drops this to just over 60% of the original reads. The final drop off shown, from millions to hundreds of sequences, is to illustrate switching from counting reads to counting unique sequences (ASV) as a tally table. The samples are the synthetic control, biological mocks, and then numerical codes for environmental samples.

The ASV sample tally table is output as a plain text tab-separated variable (TSV) file, and optionally in the Biological Observation Matrix (BIOM) format facilitating use with alternative classifiers (*McDonald et al., 2012*).

## Abundance thresholds

There are two compelling reasons to impose abundance thresholds. Firstly, most rare ASVs including singletons are generated *via* errors in either the PCR amplification or sequencing steps (*Edgar (2016)*; Fig. 4), and their removal improves the signal to noise ratio and results in a marked improvement in computational efficiency. Secondly it plays a key role in dealing with cross-sample contamination, including Illumina tag-switching (*Schnell, Bohmann & Gilbert, 2015*).

The tool implements both an absolute minimum abundance threshold defaulting to 100 copies (based on examination of our own datasets), and a fractional threshold defaulting to the widely used value of 0.1% (*Muri et al., 2020*) of the paired reads in a sample which passed primer trimming for that marker. These are applied to each sample. The fractional threshold is more appropriate than an absolute threshold if the sampling depth varies dramatically between samples. The default absolute threshold may be too high for low yield runs like the Illumina Nano Kit protocol, or if the focus is maximising sensitivity. An ASV supported by a single read is known as a singleton, and for efficiency these are always automatically excluded. In most cases singletons are a single base pair away from a more dominant sequence, and are presumed to originate from amplification or sequencing

errors, resulting in a halo effect when visualised as an edit-graph (see Fig. 3). In such cases, read correction would map them to that central node, but this is not always clear cut as there can be multiple high abundance high occurrence adjacent nodes. Unlike the tools DADA2 (*Callahan et al., 2016*), obiclean (*De Barba et al., 2014*; *Boyer et al., 2016*), or UNOISE2 (*Edgar, 2016*), we default to simply excluding these reads *via* the abundance threshold.

Another source of unwanted low abundance sequences comes from Illumina tag-switching (*Schnell, Bohmann & Gilbert, 2015*). Using metabarcoding synthetic controls, *Palmer et al. (2018)* quantified the effective rate as under 0.02%, consistent with between 0.01% and 0.03% of reads in earlier work reviewed by *Deiner et al. (2017)*. However, while excluding low abundance variants from PCR noise and tag-switching is important, as in *Muri et al. (2020)* we use a higher default of 0.1% for excluding most contamination. The tool supports a data-driven minimum abundance threshold using (unwanted) amplification in negative control samples, a widely used strategy (*Sepulveda et al., 2020*). The control samples are processed before the non-controls, in order to infer and apply a potentially higher control-driven threshold to the other samples in that batch. Sample batches are defined by providing input data in sub-folders, which could be MiSeq runs, or reflect samples amplified together.

Simple blank negative controls should contain no sequences at all, so the highest abundance sequence present can be used as an inferred absolute abundance threshold (if higher than the default), and applied to all the samples in that batch. *Massart et al. (2022)* caution however that trace levels of DNA in an otherwise empty control may amplify very efficiently, overestimating contamination, and so recommend a spike-in or positive control approach.

If the experiment uses synthetic sequences spiked into a negative control, it is possible to distinguish the expected spike-in sequences (subject only to technical noise and artifacts from PCR and sequencing) from biological contamination from laboratory practices (*Palmer et al., 2018*). In principle a biological out-group or "alien control" could be used as the spike-in (*Massart et al., 2022*), but unique novel synthetic control sequences will provide the greatest confidence. Provided the tool can identify and thus ignore the spike-in sequences, any remaining reads in those controls can be used to raise the absolute threshold. Furthermore, the percentage of the most abundant non-spike-in sequence can be taken as an inferred fractional abundance threshold (if higher than the default). *Palmer et al. (2018)* takes a more optimistic approach in their tool AMPtk by applying ASV specific thresholds, assuming the other biological sequences not observed as cross contaminants are well behaved. THAPBI PICT takes the more pessimistic approach of taking the worse case as indicative of cross contamination rates for that sample batch in general.

In our own data, (cross-)sample contamination appears to be more of an issue than Illumina tag-switching. At the time of writing we have completed 30 *Phytophthora* ITS1 MiSeq sequencing runs with synthetic control samples, covering plant nurseries (*Green et al., 2021*) and environmental samples. One run was discarded after finding 1–5% non-synthetic reads in all the controls, traced to contamination of the PCR master mix.

Another problematic run saw four of the six controls in a 96-sample amplification plate with over 2% non-synthetic reads. These had an identical ASV profile, suggesting a single contamination event after pipetting the first two controls. The dominant contaminant here was a rare *Phytophthora* species not seen on the rest of the samples being sequenced, making the most likely contamination source DNA from an older sample previously processed in the laboratory. *Thalinger et al. (2021)* has a number of recommendations on the laboratory side for minimising contamination events. By using the worst control non-synthetic read fractions as thresholds for those plates we reduce the chances of false positives, at the cost of false negatives for minority community members. This is not unprecedented-for example guided by their mock community controls, *Hänfling et al. (2016)* used thresholds of 0.3% and 1% for their 12S and cytB amplicons respectively (and an absolute threshold of at least three reads per ASV).

# CLASSIFIERS AND DATABASES

## Classifier implementations

All of the classifiers in THAPBI PICT are based on independent comparisons of each ASV to the sequences in the database as strings of letters. There is no clustering, meaning the classification can be performed on a global listing of all ASV, without considering the context of what other sequences were present in the same samples.

Technically the `identity` classifier does the matching with an SQL query within SQLite. For performance the `substr` classifier is done in Python after loading all the database sequences into memory. The edit distance based classifiers also load all the sequences into memory, and then use the Levenshtein metric as implemented in the Rapid Fuzz library (*Bachmann et al., 2022*), where a one base-pair insertion, deletion, or substitution is considered an edit distance of one. All our distance classifiers accept a species level match at most one base pair away, equivalent to about a 99.5% identity threshold (assuming amplicons around 200 bp long). This may seem high, but historic thresholds like 97% for the 16S marker are too relaxed (*Edgar, 2018*). The least stringent classifier currently implemented (`blast`) assigns the species of the best BLAST nucleotide match within the database *Camacho et al. (2009)*, ranked by bit-score subject to a minimum alignment length and score intended to exclude the most obvious false positives. In objective assessment (see below), this does over-predict (assigning sometimes tenuous species matches). This BLAST based classifier should only be used for preliminary analyses like exploring a new dataset with an uncurated database.

## Database and classifier interactions

The tool has been designed as a framework which can be applied to multiple biological contexts, demonstrated in the worked examples discussed below. In each case, a relevant reference database will need to be compiled.

Applied to environmental samples, some primer pairs will amplify a much wider sequence space than others, either reflecting a more diverse genome region, or simply from having longer amplicons. Related to this, the fraction of observed sequences with a published reference will also vary, a problem particularly in understudied organisms, or

with novel barcoding amplicons. This means the density of the references in experimentally observed sequence space is context dependent, and thus so is the most appropriate classifier algorithm.

The default classifier allows perfect matches, or a single base pair (bp) difference (substitution, insertion or deletion). This requires good database coverage with unambiguous sequences trimmed to the amplicon only, which we have been able to achieve for the *Phytophthora* ITS1 region targeted. This classifier can still be used with reference sequences containing a single IUPAC ambiguity code (which will count as the single allowed mismatch), but more than that and the reference could only be used with a less stringent classifier (such as the best BLAST nucleotide match).

## Default ITS1 database and conflict resolution

Our chosen ITS1 primers target a region of eukaryote genomes between the 18S and 5.8S rRNA genes, with nested PCR primers to selectively target *Phytophthora* (*Scibetta et al., 2012*), related paraphyletic genera of downy mildews and the sister taxa *Nothophytophthora*. They have been observed to occasionally amplify related genera, such as *Pythium* and *Phytopythium*, especially when *Phytophthora* levels in the sample are very low. Our curated database initially focused on *Phytophthora*, building on the work in *Català, Pérez-Sierra & Abad-Campos (2015)* and *Riddell et al. (2019)*. Published ITS1 sequences are often truncated to the start of the ITS1 region, and thus omit our left primer and the highly conserved 32 bp section of the 18S region at the start of our amplicon of interest, which handicapped building a reference set. In addition to using public sequences, we also performed additional Sanger capillary sequencing. Also, given that *Phytophthora* rRNA is known to be present in variable numbers of copies in a tandem array with potential variability between copies, we also ran some single isolates from culture collections through the MiSeq pipeline which determined that many species were uniform but others revealed secondary ITS1 variants. The primary goal was classification of the genus *Phytophthora*, but widening coverage to downy mildews and related genera such as *Nothophytophthora* and the rarely amplified *Pythium* created two additional challenges. First, there are fewer published sequences available, and thus the default classifier becomes too strict to assign many species. The Phyto-Threats project therefore uses a more relaxed classifier which falls back on a genus level classification based on the closest database entries up to 3 bp edits away. Second, the taxonomic annotation becomes less consistent, particularly within the former *Pythium* genus that was subject to taxonomic revision that generated new genera such as *Globisporangium* or *Phytopythium*. This led to many conflicts with database accessions of (near) identical ITS1 sequences having different genus names. These direct conflicts, and similar cases of apparent misannotation, were resolved manually by excluding the unwanted accessions in the database build script.

With any amplicon marker, it is possible that distinct species will share the exact same sequence. For example, this happens with model organism *Phytophthora infestans* and sister species such as *P. andina* and *P. ipomoeae*. In such cases the classifier reports *all* equally valid taxonomic assignments. The database author could instead record a single assignment like *Phytophthora infestans*-complex. Conversely, some *Phytophthora*

genomes are known to contain multiple copies of our target marker ITS1 through tandem repeats of the rDNA ITS region. In such cases the recognised variant forms should be added to the reference database. Despite their shortcomings, the ITS1 region has remained the de-facto standard within the oomycete community (*Robideau et al., 2011*), but alternatives are being explored (*Foster et al., 2022*).

## CLASSIFICATION ASSESSMENT

In assessing classification performance, it is the combination of both classification method (algorithm) and marker database which matters. Settings like the abundance threshold are also important, and the tool default settings partly reflect one of the original project goals being to avoid false positives.

To objectively assess a metabarcoding classifier we require sequenced samples of known composition, which generally means single isolates (where a single marker sequence is typically expected), or mock communities of known species (the bulk of our examples). Carefully controlled environmental samples may also be used, such as *Muri et al. (2020)* in our worked examples. Here a lake was drained to collect and identify all the individual fish, but this is problematic as the lake was large enough that DNA from each fish could not be expected at all the sampling points, giving an inflated false negative count.

Our tool includes a presence/absence based assessment framework based on supplying expected species lists for control samples, from which the standard true positive (TP), false positive (FP), true negative (TN), and false negative (FN) counts can be computed for each species. These are the basis of standard metrics like sensitivity (recall), specificity, precision, F-score (F-measure, or F1), and Hamming Loss. It is simple but not overly helpful to apply metrics like this to each species, rather the overall performance is more informative.

However, some scores like the Hamming Loss are fragile with regards to the TN count when comparing databases. The Hamming Loss is given by the total number of mis-predicted class entries divided by the number of class-level predictions, thus $(FP + FN)/(TP + FP + FN + TN)$. Consider a mock community of 10 species, where the classifier made 11 predictions which break down as 9 TP and 2 FP, meaning $10 - 9 = 1$ FN. Suppose the database had a hundred species (including all ten in the mock community), that leaves $100 - 9 - 1 - 2 = 88$ TN, and a Hamming Loss of $3/100 = 0.03$. Now suppose the database was extended with additional references not present in this mock community, perhaps expanding from European *Phytophthora* species to include distinct entries for tropical species, or a sister group like *Peronospora*. The denominator would increase, reducing the Hamming Loss, but intuitively the classifier performance on this mock community has not changed. To address this, the classifier assessment also includes a modified *ad hoc* loss metric calculated as the total number of mis-predicted class entries divided by the number of class-level predictions ignoring TN, or $(FP + FN)/(TP + FP + FN)$ which in this example would give $3/12 = 0.25$ regardless of the number of species in the database. This is an intuitive measure weighting FP and FN equally (smaller is better, zero is perfect), a potential complement to the F-score.

**Table 2 Species level classifier assessment on the *Riddell et al. (2019)* mock communities, with TP and FP counts from their Table 1, and FN counts from their text.** THAPBI PICT using default settings has an abundance threshold of 100 reads, also shown using just 50 reads. The theoretical best assumes everything except *Phytophthora boehmeriae* could be found, and ignores that some of the ITS1 amplicons are ambiguous at species level. F1 score or F-measure calculated as $2TP/(2TP + FP + FN)$, given to 2 dp. *Ad hoc* loss defined as $(FP + FN)/(TP + FP + FN)$, given to 3 dp.

| Mock community | Method | TP | FP | FN | F1 | *Ad hoc* loss |
|---|---|---|---|---|---|---|
| 15 Species mix | Metapy/bowtie | 11 | 1 | 4 | 0.81 | 0.333 |
| 15 Species mix | Metapy/swarm | 14 | 4 | 1 | 0.85 | 0.263 |
| 15 Species mix | THAPBI PICT (defaults) | 8 | 2 | 7 | 0.64 | 0.529 |
| 15 Species mix | THAPBI PICT (50 reads) | 11 | 3 | 4 | 0.76 | 0.389 |
| 15 Species mix | Theoretical best | 14 | 0 | 1 | 0.97 | 0.067 |
| 10 Species mix | Metapy/bowtie | 7 | 6 | 3 | 0.61 | 0.563 |
| 10 Species mix | Metapy/swarm | 9 | 10 | 1 | 0.62 | 0.550 |
| 10 Species mix | THAPBI PICT (defaults) | 8 | 2 | 2 | 0.80 | 0.333 |
| 10 Species mix | THAPBI PICT (50 reads) | 8 | 2 | 2 | 0.80 | 0.333 |
| 10 Species mix | Theoretical best | 9 | 0 | 1 | 0.95 | 0.100 |
| Combined | Metapy/bowtie | 18 | 7 | 7 | 0.72 | 0.438 |
| Combined | Metapy/swarm | 23 | 14 | 2 | 0.74 | 0.410 |
| Combined | THAPBI PICT (defaults) | 16 | 4 | 9 | 0.71 | 0.448 |
| Combined | THAPBI PICT (50 reads) | 19 | 5 | 6 | 0.78 | 0.367 |
| Combined | Theoretical best | 23 | 0 | 2 | 0.96 | 0.080 |

Note that the assessment framework only considers species level predictions, ignoring genus only predictions and unknowns, and thus will not distinguish between the default `onebp` classifier and variants like `1s3g` (see Table 1).

As a benchmark of the default classifier and *Phytophthora* focused database, we used the 10 and 15 species mixes in *Riddell et al. (2019)*, see Table 2. This was originally analysed with the `metapy` pipeline with a high stringency classifier using `bowtie` to find perfect alignments, and a more relaxed classifier using `swarm` for clustering. In both samples and both classifiers, *Phytophthora boehmeriae* was not found, and this was attributed to uncompetitive amplification in a mixed DNA sample due to poor PCR primer binding. That being so, the best classifier results would be either 14 TP and 9 TP respectively, with 0 FP if the markers were unique, and 1 FN. However, note that not all the markers are unique, both mixes contain species known to share their ITS1 marker with other species, giving unavoidable technical FP, also discussed in *Riddell et al. (2019)*.

Using F1 score or our *ad hoc* loss ranking, THAPBI PICT is clearly performing best on the 10 species mix (and better than metapy did). However, with default settings it does worse on the 15 species mix, due the high FN count where the default ASV abundance threshold of 100 reads is excluding expected species. In this MiSeq dataset the Illumina Nano Kit was used giving lower yields, making the default 100 read threshold overly harsh. Optimising on maximising the F1 score and minimising ad-hoc-loss, and weighting the two communities equally, suggests running THAPBI PICT with an ASV
read abundance threshold of around 50 reads performs best overall here. This is a fundamental problem however, low abundance community members can be indistinguishable from background noise/contamination, meaning without controls the best threshold is arbitrary.

## REPORTING

The pipeline produces two tabular reports (which can also be requested directly with the `summary` command), output as both tab-separated plain text, and Excel format with colouring and conditional formatting (Fig. 2). These include information on read counts from the preparation stage (as used in Fig. 4), information on the abundance thresholds, and foremost the species classification from the chosen method. The user may provide a table of metadata cross referenced by the sample FASTQ filename stem, which will be used for sorting the samples and if possible colouring inferred sample groupings (*e.g.*, sample source, or replicates) to ease interpretation. This allows quick visual comparison of replicates as adjacent rows/columns.

The read report by default sorts the ASVs by their taxonomic classification, and then by abundance. This makes it easy to identify the most common unknowns or genus-only predictions for manual review (using the ASV sequence). This sorting also means that when the thresholds are low enough to let through noise, the grey halo effect shown in the edit graph (see Fig. 3) is also visually distinct as highly abundant rows followed by less abundance variants. This read report can also be exported in BIOM format.

For many of the worked examples the sample metadata on the NCBI Short Read Archive (SRA) or European Nucleotide Archive (ENA) had to be supplemented by information in the associated publication. Providing such metadata to the archives using an approved ontology based checklist is non-trivial, but adds greatly to the reuse potential (*Tedersoo et al., 2015*). We provide an `ena-submit` command which facilitates using the interactive ENA upload step for matching FASTQ filenames to previously entered sample information.

The tool's repository includes a number of helper scripts, including a pooling script written for the Phyto-Threats project for preparing plant nursery specific summary reports. This simplifies the sample report by combining replicate samples into a single row, and can either use the read count sum, or just "Y" (present) or "N" (absent).

The other noteworthy report from the tool is an edit graph, invoked *via* the `edit-graph` command, as shown in Fig. 3. By default this outputs the edit graph in XGMML format which can then be visualised in a tool like Cytoscape (*Shannon et al., 2003*), with a choice of node layouts and representations (*e.g.*, customising node size by sample count, or colour by genus). The graph can help guide the choice of minimum abundance threshold (as discussed above), and the choice of classifier. In the example shown with a 3 bp maximum edit-distance shown, the cliques formed are for the most part clearly distinct species, with a single central node. With the default ITS1 marker used for *Phytophthora* we find greater sequence variation and therefore more diverse non-simple clusters for species like *Phytophthora nicotianae* and *P. gonapodyides*, but most species show a single central ITS1 sequence.

## DEVELOPMENT PRACTICES

THAPBI PICT is released as open source software under the MIT licence. It is written in Python, a free open source language available on all major operating systems. Version control using git hosted publicly on GitHub at https://github.com/peterjc/thapbi-pict/ is used for the source code, documentation, and database builds including tracking the hand curated reference set of *Phytophthora* related ITS1 sequences. Continuous integration of the test suite is currently run on both CircleCI (Linux) and AppVeyor (Windows). Software releases are to the Python Packaging Index (PyPI) as standard for the Python ecosystem, and additionally packaged for Conda *via* the BioConda channel (*Grüning et al., 2018*). This offers simple installation of the tool itself and all the command line dependencies on Linux or macOS. Installation on Windows requires manual installation of some dependencies. The documentation is currently hosted on Read The Docs, updated automatically from the GitHub repository.

## WORKED EXAMPLES

In this section we briefly discuss the application of THAPBI PICT to public data sets from several published articles, covering a range of organisms and markers. The selection has prioritised examples including mock communities and negative controls, and have been included in the tool documentation as worked examples. These worked examples generally are highly concordant with the published analyses, with differences largely down to the exact choice of thresholds.

The example scripts first-run times range from a few minutes with under 1 GB of raw FASTQ data (*Bakker, 2018*; *Riddell et al., 2019*; *Walker et al., 2019*; *Muri et al., 2020*), to a few hours with the larger datasets like *Ahmed et al. (2019)* with 12 GB of input. These times are dominated by the merging the paired reads during read preparation stage, and as discussed earlier, running the read preparation stage in parallel across a cluster is advised on larger projects.

The first worked example is a simple one using the provided *Phytophthora* ITS1 database we have generated for this work to reexamine *Riddell et al. (2019)*. This example does not include the synthetic controls introduced later, but does have blanks as negative controls and simple mock communities as DNA mixes (discussed above for classifier assessment). The second example uses *Redekar, Eberhart & Parke (2019)* but focuses on how to build a database, including how species names can optionally be validated against the NCBI taxonomy.

The example based on *Muri et al. (2020)* is a single 12S marker for fish, with a custom database including numerous off-target matches like humans and sheep. In this case the lake contents were determined by draining the lake and collecting the fish, but this did not determine which of the sampling sites any given fish might have visited, complicating interpretation compared to an artificial mock community. Another single marker example based on *Walker et al. (2019)* uses COI in simple mock communities of bats, and shows the importance of the database content with the default classifier. The most interesting single marker example is based on *Palmer et al. (2018)*, fungal ITS2 markers with mock biological

fungal communities and *synthetic control* sequences. This has been discussed above in the context of setting abundance thresholds.

There are examples with multiple markers which were sequenced *separately* in *Klymus, Marshall & Stepien (2017)*, two different 16S mitochondrial markers with mock communities, and *Ahmed et al. (2019)*, four different markers in mock nematode communities. The example in *Batovska et al. (2021)* uses three markers together, while *Arulandhu et al. (2017)* sequences over a dozen markers together. Here the primer sequences themselves are non-overlapping and so serve to separate out the amplicons for each sample, allowing them to be matched to the relevant reference set. Note currently a primer cocktail as used for the COI example in this data set is not supported. This article is also noteworthy as an inter-laboratory replication study of metabarcoding.

Datasets from some potentially useful publications could not be used directly, generally for technical reasons. Many used custom multiplexing (*Elbrecht & Leese, 2015*; *Port et al., 2016*; *Elbrecht et al., 2016*, *2019*; *Elbrecht, Peinert & Leese, 2017*; *Braukmann et al., 2019*), and thus would require separate de-multiplexing before use. Some like *Braukmann et al. (2019)* and *Duke & Burton (2020)*, use an amplicon too long to span with overlapping Illumina MiSeq paired reads. Sometimes, however, articles did not provide the *raw* FASTQ files. For instance, *Blanckenhorn et al. (2016)* did not share the raw FASTQ files at all, while *Hänfling et al. (2016)* and *Zaiko et al. (2022)* provided primer trimmed FASTQ files. Some older articles (also) used the Roche 454 or Ion Torrent platforms, which would require re-engineering mainly around the different error profile, which is potentially unsuited for our default strict classifier.

## DISCUSSION

Here we present a novel and flexible pipeline for the objective analysis of metabarcode data, not just of single markers but also pooled markers where the amplicons can be separated *via* their primer sequences. Some of the design choices and default settings reflect the initial use case being *Phytophthora* ITS1 sequence markers in a context where specificity was favoured over sensitivity. In general, appropriate abundance thresholds and classifier algorithm will be experiment and/or marker specific, with the quality of the reference database a key factor. All amplicon barcoding experiments should be designed with suitable controls to assess the limits of quantification *vs* presence/absence (*Lamb et al., 2019*), including the effects of the PCR (*Thielecke et al., 2017*) and contamination (*Thalinger et al., 2021*).

By design, the tool currently reports lists of genus/species for each ASV, without attempting anything like a most recent common ancestor analysis. This limitation can be a handicap with some use-cases where the markers may not readily resolve at species level, and/or an ASV is often shared between genera. See for example, the Brassicaceae discussed in *Arulandhu et al. (2017)*, and fish examples in *Muri et al. (2020)*. Moreover, it makes the tool unsuited to markers like regions of the bacterial 16S rRNA gene which are typically used at phylum level with environmental datasets (*Straub et al., 2020*). Rather it is appropriate for comprehensive analyses of better defined taxonomic markers such as the plant pathogenic oomycete ITS1 marker used primarily for *Phytophthora*, where it is

proving valuable for the ongoing characterisation of a comprehensive set of several thousand samples from plant nurseries in the Phyto-Threats project (*Green et al., 2021*) and in natural ecosystems (*Riddell et al., 2019*).

Our pipeline supports using negative or synthetic spike-in controls to set an abundance threshold on groups of samples (such as each sequencing run). Rather than ASV-specific thresholds as in *Palmer et al. (2018)*, THAPBI PICT takes the more cautious approach of interpreting the worst case as indicative of cross contamination rates for that sample batch in general.

The pipeline does not currently explicitly attempt to find and remove chimera sequences beyond the use of abundance filters. As discussed in *Edgar (2016)*, chimeras which are also only 1 bp away from a reference sequence cannot be distinguished from a point error, and would be matched to that reference by all but our strictest identity classifier. Apart from this special case, any high abundance chimera would likely appear in our reports as an unknown, and would most likely be only in a single sample. Regular manual inspection of the high abundance unknown reads appearing in multiple samples was part of the ongoing quality assurance during the Phyto-Threats project, primarily to identify any gaps in the database coverage. The only clear chimeras identified were from our synthetic controls, where part of our non-biological sequence was fused to some unexpected sequence. Potentially more complex mock communities of synthetic sequences could be used to generate a gold standard for identifying chimeras which might serve as a benchmark dataset for testing chimera algorithms.

Another important difference from other ASV based tools like DADA2 (*Callahan et al., 2016*), obitools (*Boyer et al., 2016*) and UNOISE2 (*Edgar, 2016*), is THAPBI PICT does not by default attempt read correction. From the halo pattern of PCR induced variants seen from synthetic inputs as viewed on an edit-graph, like Fig. 4B, there is usually a natural central node to which a variant can be attributed. However, the situation is not always clear cut, with some species like *Phytophthora gonapodyides* showing a range of known ITS1 sequences. Rather our approach is to exclude most PCR noise through the abundance filters, and allow a modest amount of variation when matching the higher abundance sequences to the reference database. As an option, however, the pipeline can apply our re-implementation of the original UNOISE2 Levenstein distance based read-correction described in *Edgar (2016)*, or invoke the UNOISE3 algorithm implemented in the USEARCH or VSEARCH tools. Read-correction seems most appropriate where the reference sequences are well separated, unlike our default *Phytophthora* ITS1 amplicon where a single base pair can distinguish known species, and thus read correction can mask lower abundance species under their more abundant neighbours.

Examination of mock community samples of our synthetic spike-in sequences showed ASV abundance to be at best semi-quantitative, as found in other work (*Palmer et al., 2018*; *Lamb et al., 2019*). For the Phyto-Threats project reports sent to plant nursery owners, we therefore only use species presence/absence (above or below the abundance threshold, and pooled replicates). However, the raw abundances are in the main tool reports, and can be used for plots or a quantitative interpretation where appropriate. The nested primer

protocol with two rounds of PCR may be a factor in undermining quantitative interpretation, and increasing the risk of cross-sample or other sample contamination.

## CONCLUSION

Here we present a novel and flexible pipeline for the objective analysis of metabarcode data, with user friendly reports including ASV read counts enabling custom graphs, as well as summary species lists per sample. While initially designed for *Phytophthora* ITS1 sequence markers, the THAPBI PICT tool can be applied more generally, including to samples containing multiple marker regions. It is best suited to markers where ASV are at least genus specific. Care should be taken picking appropriate abundance thresholds, which can be set using negative and/or synthetic controls, and in applying read-correction for de-noising. While high-throughput amplicon sequencing does give read counts per species (or per ASV), we and others caution against treating these as quantitative (*Palmer et al., 2018*; *Lamb et al., 2019*). The most suitable classifier algorithm will be marker specific, with the quality and coverage of the reference database a key factor. Including mock communities in your experiment allows the performance of classifier and database to be evaluated objectively.

## ACKNOWLEDGEMENTS

We thank our colleagues at the James Hutton Institute including Eva Randall, Beatrix Keillor, Pete Hedley, and Jenny Morris, and at Forest Research including Sarah Green, Debra Frederickson Matika and Carolyn Riddell. The reviewers are thanked for their constructive feedback, Celine Mercier, Alexander Piper, and the anonymous Reviewer 3. The authors acknowledge the Research/Scientific Computing teams at The James Hutton Institute and NIAB for providing computational resources and technical support for the "UK's Crop Diversity Bioinformatics HPC" (BBSRC grant BB/S019669/1), use of which has contributed to the results reported within this article.

### Funding

This research was supported by a grant funded jointly by the Biotechnology and Biological Sciences Research Council (BBSRC), Department for Environment, Food and Rural affairs (DEFRA), Economic and Social Research Council (ESRC), Forestry Commission, Natural Environment Research Council (NERC) and Scottish Government, under the Tree Health and Plant Biosecurity Initiative, grant number BB/N023463/1. Also partly funded by DEFRA as part of the Future Proofing Plant Health project in support of Euphresco ID-PHYT, and by the Rural & Environment Science & Analytical Services (RESAS) Division of the Scottish Government. The funders had no role in study design, data collection and analysis, decision to publish, or preparation of the manuscript.

## Grant Disclosures

The following grant information was disclosed by the authors:

Biotechnology and Biological Sciences Research Council (BBSRC).

Department for Environment, Food and Rural affairs (DEFRA).

Economic and Social Research Council (ESRC).

Forestry Commission, Natural Environment Research Council (NERC).

Scottish Government, under the Tree Health and Plant Biosecurity Initiative: BB/N023463/1.

DEFRA.

Euphresco ID-PHYT.

Rural & Environment Science & Analytical Services (RESAS).

Division of the Scottish Government.

## Competing Interests

The authors declare that they have no competing interests.

## Author Contributions

- Peter J. A. Cock conceived and designed the experiments, performed the experiments, analyzed the data, prepared figures and/or tables, authored or reviewed drafts of the article, and approved the final draft.
- David E. L. Cooke conceived and designed the experiments, analyzed the data, authored or reviewed drafts of the article, and approved the final draft.
- Peter Thorpe conceived and designed the experiments, performed the experiments, authored or reviewed drafts of the article, and approved the final draft.
- Leighton Pritchard conceived and designed the experiments, analyzed the data, authored or reviewed drafts of the article, and approved the final draft.

## Data Availability

The source code and documentation are available at GitHub and Zenodo: https://github.com/peterjc/thapbi-pict/.

Peter Cock, Leighton Pritchard, Peter Thorpe, & David Cooke. (2023). peterjc/thapbi-pict: THAPBI PICT v1.0.0 (v1.0.0). Zenodo. https://doi.org/10.5281/zenodo.7950664.

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
