# Peer review of "THAPBI PICT—a fast, cautious, and accurate metabarcoding analysis pipeline"

_PeerJ, doi:10.7717/peerj.15648_

## Round 0.1 · original submission · Minor Revisions

Dear Peter and co-authors,

I have received three independent reviews of your study. While all reviewers clearly recognised the quality/novelty of your work, they have collectively raised a number of issues that will need to be addressed in your revised manuscript.

Overall, the reviewers have provided you with excellent suggestions on how to improve the manuscript, and I will be looking forward to receiving your revised manuscript along with a point-by-point response to their comments.

With warm regards,
Xavier

·

Basic reporting

1. Overall the manuscript is written in clear and correct English, but there is a significant amount of missing words and letters.
Please fix lines: 125-126, 138, 234, 245, 266, 319, 322-323, 349, 358.

2. The introduction presents the context and defines the problem clearly (except for the definition of ‘multiplexed’, see point 6).

3. The references are appropriate. Maybe add some references for the statements made lines 137-141.

4. The structure of the article does not conform to the standard format, but can be justified by the fact that the article describes a bioinformatics pipeline.

5. The “CLASSIFIERS” and “DATABASE AND CLASSIFIERS” sections should maybe be merged together, and divided into more subsections.

Abstract, line-specific comments:

6. L 13-14: “for metabarcoding analysis with multiplexed Illumina paired-end reads, including where different amplicons are sequenced together” is redundant, unless the authors’ definition of “multiplexed” is not “multiple markers sequenced together”. The authors should maybe define what they mean with “multiplexed” in the introduction.

7. L 16: Maybe add a sentence about how the pipeline was initially designed for Phytophthora detection before “THAPBI PICT can be applied to other amplicons and organisms”.

Figures and tables: Overall OK.

8. Figure 1: Typo in “In additional BIOM format output can be requested”.

9. Figure 2: “from one of the MiSeq runs in Palmer et al. (2018)”: Specify which one if possible.

10. Figure 3: Describe what the red colour represents for nodes.

11. Figure 4: The text is a bit small but can be read when zoomed-in.

Data and software availability:

12. There is a typo in the github instructions to install the software through conda. The command should be:
conda install thapbi-pict

13. Version 0.14.1 was installed and tested on one of the examples provided through the github repository. The code was briefly checked.

Experimental design

14. The research is within the scope of the journal (bioinformatics tool).

15. Research originality: the authors initially developed the pipeline to be tailored for the detection of Phytophthora species, with some associated specific assumptions such as: the target sequences are amplified in decent quantities if present in a sample ; the target sequences are potentially highly similar, and need to be differentiated ; and close matches are represented in the reference databases at the species or genus level. More options are offered for experiments with different properties.

16. The methods are described with sufficient details and information to replicate. The investigations were conducted rigorously.

Validity of the findings

17. Line 202: “All our distance classifiers accept a species level match at most one base pair away, equivalent to about a 99.5% identity threshold.”: Specify that this is only true for markers of about 200bp (or remove the equivalence statement).

18. The choices made regarding abundance thresholds and other filtering steps are appropriately discussed and justified, especially in relation to the initial purpose of the pipeline.

19. The classification assessment is correctly done, described, and discussed.

20. The conclusion addresses the main specificities of the pipeline appropriately.

Additional comments

21. The described pipeline is mostly suited for the detection of specific, well-described species (i.e. with reference sequences at the species or genus level), and not for e.g. biodiversity analysis from huge datasets of less-known taxa. This is mostly well discussed in the manuscript, except for the scaling to big databases and big datasets. In fact, the computational efficiency of the pipeline is not discussed at all in the manuscript. It could be interesting to describe and discuss it, even briefly.

·

Basic reporting

In this article Cock and colleagues present a novel open-source bioinformatic pipeline for analysis of metabarcoding datasets, including extensive validation to confirm its accuracy and applicability across various taxonomic groups and barcode markers. The article describing this software is very well written, and provides a comprehensive overview of the pipeline and relevant outputs, as well as the process of validation, and technical information on the software dependencies and development practices involved.

Experimental design

This pipeline makes some variations compared to other widely-used alternatives (i.e., the default settings omit clustering or denoising in favour of minimum abundance thresholds, and use edit-distance based taxonomic assignment) to increase the speed of analysis. These adaptations from what is currently the norm are well justified in the text and validated using publicly available and newly generated test datasets. The validation of the pipeline is well designed, covers multiple taxonomic groups and markers, and the metrics used to quantify performance are appropriate.

Validity of the findings

All the results of the pipeline validation are clearly presented and discussed in the context of the literature. In addition to the article there is extensive online documentation provided which covers the installation and use of the software, including the data and code for a set of worked examples covering different target taxonomic groups and barcode markers. From this documentation it is clear that a lot of effort has gone into making this software pipeline accessible to new users and I commend the authors for this. As part of this review, I tested the software following one of these worked examples and found the software relatively simple to install and use, with the documentation easy to follow. The robust software development practices that are described in the article alongside the extensive documentation gives me confidence that this software will be well maintained into the future.

Additional comments

While I have made a few very minor comments below, I believe this manuscript and software pipeline would be a great addition to the literature in its current state. Overall, I believe this to be a comprehensive and well validated metabarcoding analysis pipeline and well written paper which has relevance to researchers and managers wishing to implement high-throughput metabarcoding for biosecurity or biodiversity assessment.

Minor comments:

Line 166: The approach of using the highest abundance sequence in the negative controls as an absolute abundance threshold can in some cases be overly conservative, as amplification is generally more efficient in negative controls due the absence of other DNA in the sample, which could risk overestimating the contamination level. It might be worth mentioning this risk here. See discussion of negative controls in Massart et al 2022 - Guidelines for the reliable use of high throughput sequencing technologies to detect plant pathogens and pests

Line 234: Remove the “additional to” in “In addition to using public sequences, we also performed additional to Sanger capillary sequencing”.

Line 294: Should this be “competitive” rather than “noncompetitive” amplification? From my understanding the amplicons from each species are competing during the PCR, with the lower efficiency amplicon (due to poor primer binding) dropping out due to competition for reagents / sequencing reads with the higher efficiency amplicons.

Reviewer 3 ·

Basic reporting

The manuscript text is written excellently and covers all eventualities and potential pitfalls.

Experimental design

Original primary research: Yes
Research question well defined, relevant & meaningful: Yes
It is stated how research fills an identified knowledge gap: for use-case Yes, universal appliaction: No
Rigorous investigation performed to a high technical & ethical standard: Yes
Methods described with sufficient detail & information to replicate: No

I encourage the authors to include the standard parameters used for the implemented tools. Especially Illumina read joining is sensitive to parameter choice.

Validity of the findings

Impact: Low
novelty: Yes
All underlying data have been provided; they are robust, statistically sound, & controlled: Yes
Conclusions are well stated, linked to original research question & limited to supporting results: In part

The conclusions reach far beyond the stated results. There is no doubt the presented pipeline works extremely well for the purpose it was developed for. The experimental design and the findings are well-supported for the described use-case.
I strongly dispute the claims of universality and would encourage to re-phrase this statement (at least to the kingdom of fungi).

Additional comments

There is a disconnect between the manuscript text and the actual implemented pipeline.
While the manuscript text is written excellently and covers all eventualities and potential pitfalls, but the actual implementation of the tool is lacking in universality and struggles to keep up with modern standards. It is puzzling why in 2023 the decision was made to implement a Levenshtein edit distance as a classification algorithm when there are k-mer and machine learning algorithms available which, in some cases, even implement a LCA approach. The arbitrary threshold selected for the blast classification lacks any justification or discussion.
The classification algorithm ignores the fact that barcodes can be shared between species because it does not output a list of the (multiple) top-hits.

The background removal approach can be identified to have stemmed from a lab with very high standards and will not behave well in field laboratories. An alternative implementation of the mentioned ASV-specific threshold, and/or keeping "contaminant" reads/ASV until after the taxonomic classification, would have allowed a much more universal application of the pipeline.

The fact hurting the study the most is the fact that the widely used DADA2 is mentioned, but not used for any kind of comparison. For this reason, it is unclear to me which knowledge gap is being filled here. It just seems to be an alternative approach without clearly demonstrating its own strengths over other metabarcoding pipelines because the manuscript lacks any kind of comparisons backed up by data.

---

## Round 0.2 · accepted · Accept

Dear Authors,

I am pleased to accept this revised manuscript for publication in PeerJ - Congratulations!

I also take the opportunity to thank the reviewers for their valuable contribution in improving this work.

With warm regards,
Xavier

·

Basic reporting

All points from the initial review have been appropriately addressed.

Experimental design

no comment

Validity of the findings

no comment

Reviewer 3 ·

Basic reporting

The authors have addressed all of my concerns, if any.

Experimental design

The authors have addressed all of my concerns, if any.

Validity of the findings

The authors have addressed all of my concerns, if any.

Additional comments

The authors have addressed all of my concerns, if any.